# Site Density Functional Theory and Structural Bioinformatics Analysis of the SARS-CoV Spike Protein and hACE2 Complex

**DOI:** 10.3390/molecules27030799

**Published:** 2022-01-26

**Authors:** Nitesh Kumawat, Andrejs Tucs, Soumen Bera, Gennady N. Chuev, Marat Valiev, Marina V. Fedotova, Sergey E. Kruchinin, Koji Tsuda, Adnan Sljoka, Amit Chakraborty

**Affiliations:** 1School of Mathematics, Statistics and Computational Sciences, Central University of Rajasthan, Ajmer 305817, India; 2018phdmt008@curaj.ac.in; 2Graduate School of Frontier Sciences, The University of Tokyo, Kashiwa 277-8568, Japan; a.tucs@edu.k.u-tokyo.ac.jp (A.T.); tsuda@k.u-tokyo.ac.jp (K.T.); 3Department of Microbiology, University of Tennessee, Knoxville, TN 37996, USA; soumenmath4189@gmail.com; 4Institute of Theoretical and Experimental Biophysics, Russian Academy of Sciences, 142290 Pushchino, Russia; 5Molecular Sciences Software Group, Environmental Molecular Sciences Laboratory, Pacific Northwest National Laboratory, Richland, WA 99352, USA; marat.valiev@pnl.gov; 6G.A. Krestov Institute of Solution Chemistry, Russian Academy of Sciences, 153045 Ivanovo, Russia; hebrus@mail.ru (M.V.F.); kruchininse@gmail.com (S.E.K.); 7RIKEN Center for Advanced Intelligence Project, Tokyo 103-0027, Japan; 8Research and Services Division of Materials Data and Integrated System, National Institute for Materials Science, Tsukuba 305-0044, Japan; 9Department of Chemistry, York University, Toronto, ON M3J 1P3, Canada; 10Department of Mathematics, School of Physical Sciences, Sikkim University, Gangtok 737102, India

**Keywords:** site density functional theory, 3DRISM, spike protein binding to human receptor, coronavirus, structural rigidity, normal mode analysis, conformational dynamics

## Abstract

The entry of the SARS-CoV-2, a causative agent of COVID-19, into human host cells is mediated by the SARS-CoV-2 spike (S) glycoprotein, which critically depends on the formation of complexes involving the spike protein receptor-binding domain (RBD) and the human cellular membrane receptor angiotensin-converting enzyme 2 (hACE2). Using classical site density functional theory (SDFT) and structural bioinformatics methods, we investigate binding and conformational properties of these complexes and study the overlooked role of water-mediated interactions. Analysis of the three-dimensional reference interaction site model (3DRISM) of SDFT indicates that water mediated interactions in the form of additional water bridges strongly increases the binding between SARS-CoV-2 spike protein and hACE2 compared to SARS-CoV-1-hACE2 complex. By analyzing structures of SARS-CoV-2 and SARS-CoV-1, we find that the homotrimer SARS-CoV-2 S receptor-binding domain (RBD) has expanded in size, indicating large conformational change relative to SARS-CoV-1 S protein. Protomer with the up-conformational form of RBD, which binds with hACE2, exhibits stronger intermolecular interactions at the RBD-ACE2 interface, with differential distributions and the inclusion of specific H-bonds in the CoV-2 complex. Further interface analysis has shown that interfacial water promotes and stabilizes the formation of CoV-2/hACE2 complex. This interaction causes a significant structural rigidification of the spike protein, favoring proteolytic processing of the S protein for the fusion of the viral and cellular membrane. Moreover, conformational dynamics simulations of RBD motions in SARS-CoV-2 and SARS-CoV-1 point to the role in modification of the RBD dynamics and their impact on infectivity.

## 1. Introduction

The coronavirus pandemic COVID-19, caused by Severe Acute Respiratory Syndrome coronavirus (SARS-CoV-2), continues to pose a serious threat across continents. Despite large volume of data on molecular structures of SARS-CoV-2 [1], essential details of molecular mechanism of interaction between SARS-CoV-2 and human cellular membrane receptor angiotensin-converting enzyme 2 (hACE2) are still under investigation. To better understand virus transmissibility, a large number of molecular studies have focused on the viral entry processes that are mediated by the spike glycoprotein (S protein), which is responsible for the receptor recognition and membrane fusion [2,3]. With the use of S protein the coronavirus hijacks the hACE2, which is highly expressed in lungs, heart, kidneys and intestine cells [2]. The S protein protomer is made of two subunits S1 and S2. The former unit, which comprises the receptor-binding domain (RBD), binds to the peptidase domain of hACE2 and contributes to stabilization of the prefusion conformational state. The SARS-CoV uses ectodomain trimer to mediate this viral entry [4,5]. Both SARS-CoV-1 and SARS-CoV-2 recognize hACE2 through its RBD, which is positioned within the flexible S1 unit of S-protein protomer.

Various computational studies have been conducted to understand the mechanism of viral infection and how it relates to inhibition of spike RBD–hACE2 binding. In order to have deeper understanding of this process, we will focus on applications of the density functional theory (DFT). It is an efficient tool for treating many-body phenomena in condensed matter systems. As a general method, DFT has been applied to problems in both quantum and classical domains. However, its impact of the applications to these areas is dramatically different. Advantages and merits of quantum DFT are well documented (see for example, [6,7]), whereas benefits and achievements of classical DFT are less evident. There are brilliant examples of DFT applications to simple fluids (see, for example [8]). However, the DFT treatment of molecular liquids remains a challenge. The bottleneck is that it requires evaluations of six-dimensional density profiles ρ(r,w) in the case of rigid molecules:(1)$$\rho \left(r,w\right)=\langle \sum_{s=1}^{N}\delta (r-x_{s})\delta (w-w_{i})\rangle$$
where brackets mean statistical averaging, while $x_{s}$ denotes position of solvent molecule *i* (e.g., center of mass) and $w_{i}$ its angular orientation (e.g., Euler angles). Despite of a recent progress in this field [9,10] the problem is still not solved for biological macromolecules due to the “curse of dimensionality”.

An alternative has been proposed by Chandler [11] who introduced site density representation, i.e., density of individual atoms in solvent molecules. The idea has led to the development of the site density functional theory (SDFT) based on the atomistic or site density description:(2)$$\rho_{i}\left(r\right)=\langle \sum_{s=1}^{N}\delta (r-r_{is})\rangle$$
where indices *i* and *s* refer to atomic site and molecule indices respectively. This site based view is well adapted for problems where atomic level of details is of importance. One of the main benefits of the SDFT is ability to analyze contributions down to individual atoms and treatment of chemical bonds. The latter are strong but localized interactions, which have to be considered in conjunction with softer long-range inter-molecular forces. Simultaneous rigorous treatment of these two very different interactions scales poses a significant problem, constituting a single biggest obstacle in site density models of molecular liquids. This has been recognized in the original formulation by Chandler, McCoy, Singer (CMS) [12,13] and follow-up developments [14,15,16,17,18,19,20,21]. Despite a recent progress in the SDFT treatment of simple solutes [22,23,24], its practical application to biological molecules has turned out to be difficult. Underlying reason of this difficulty is related to treatment of chemical bonds, that necessitates to perform angular averaging directly. There is a simplified SDFT version referred to as the three-dimensional reference interaction site model (3DRISM) [25,26]. It treats molecular liquids as an “effective” atomic mixture, which makes it possible to utilize many of the techniques from simple fluid theory. Namely, the 3DRISM reduces the problem to numerical solution of Ornstein-Zernike integral equations similar to those for simple fluids. The model is very popular for treatment of small bio-rganic solutes [27,28,29,30,31,32,33,34,35,36,37,38,39,40,41,42,43,44,45,46,47,48,49,50,51,52,53,54].

Despite certain success of the 3DRISM applications, only few of them are devoted to coronaviruses [55,56]. The reason for that lies not only in the large size of viruses but also a difficulty in treatment of the flexibility of coronavirus structure. Although there is a possibility to address this issue by combining 3DRISM and conventional molecular mechanics treatment [57], the computational costs of such combined method makes its applications to virus systems impractical. One possible way to overcome this bottleneck is to integrate 3DRISM with bioinformatics methods. The latter is widely used to evaluate protein-protein and protein-ligand interactions. The structural bioinformatics approach involves a wide range of tools such as molecular docking, evaluations of affinity and binding constants, constrained geometric simulations (CGS) for conformational sampling, normal mode analysis (NMA), etc. [58]. Typically, to reduce computational costs, the bionformatics approach limits molecular level description to the interacting protein and ligands, and treats environmental effects in a simplified manner (e.g., generalized Born surface area (GBSA) model [59]). The latter is accurate only when electrostatic interactions dominate. However, short ranged protein-solvent interactions involving hydrogen bond formation, steric repulsion, and Van-der-Waals attraction play an essential role in the protein-protein complexation. The 3DRISM seems to be a suitable methodological choice to reveal a role of electrostatic and short-ranged protein-solvent interactions in solvation thermodynamics, as the theory can be used to detect water-mediated contacts in protein complexes and identify favorable or unfavorable contributions to the binding free energy. As such, integration of 3DRISM calculations with structural bioinformatics treatments represents an attractive strategy. The latter may be used to select “regions of interests” that can be analyzed by 3DRISM, providing an insight into the role of water-mediated interactions. Furthermore bioinformatics analysis can help to evaluate the role of rigidity and flexibility of the complexes under consideration. The goal of this paper is to test the utility of integrated classical SDFT and bioinformatics methodology for simulating COV spike protein. In particular, we investigate formation and stability of SARS-CoV-2-hACE2 and SARS-CoV-1-hACE2 complexes. The scope of the paper is the following. First, we provide the preliminary bioinformatics treatment to reveal main characteristics of binding between the RBD domain of spike protein and hACE2 receptor (Section 2.1). Then, we analyze the role of water-mediated interactions in the formation of this complex using 3DRISM (Section 2.2). We also test how replacement of SARS-CoV-1 by SARS-CoV-2 will affect the binding of RBD domain to hACE2. To validate the results we provide further bioinformatics analysis using the NMA and CGS methods (Section 2.3). Finally, we summarize the revealed effects and discuss briefly benefits and bottleneck of the combined 3DRISM and structural bioinformatics tools. We outline the basics of the classical SDFT and the main features of the applied bioinformatics tools in Section 4. Essential details of the evaluations and additional evidences can be found in Appendix A.

## 2. Results

### 2.1. Preliminary Bioinformatics Treatment

SARS-CoV-2-S protein holds 98% sequence similarity with the bat coronavirus RaTG13. A most critical variation observed in CoV-2 is an insertion “RRAR” at the furin recognition site at S1/S2 junction, while SARS-CoV-1 has single arginine at that site. Besides this insertion, 60% residual substitutions are noted at the RBD domain [60]. Examining how such differences contribute to higher recognition capability of hACE2 receptor is important to underpin the therapeutic target that can prevent virus entry. Here we analyze respective protomers of four S-protein cryo-EM structures: SARS-CoV-1 (pdb id: 6CRZ) [61], SARS-CoV-2 (pdb id: 6VYB) [62], SARS-CoV-RBD-hACE2 complex (pdb id: 2AJF) [63], and SARS-CoV-2-RBD-hACE2 complex (pdb id: 6M17) [64]. Recent work has revealed that the S-protein in the open state with at least one RBD in the “up” conformation, corresponds to the receptor accessible conformation that can bind to hACE2. In comparison to the CoV-1-RBD up state, CoV-2-RBD has expanded its net surface area, undergoing a large local conformational change, and a relatively large amplitude anti-phase-like RBD motion in the slow-motion second normal mode. In order to recognize the host receptor, the RBD of S1 undergoes hinge-like conformational motions. In a recently reported structure of SARS-CoV-2, it was observed that hACE2 can only bind when the RBD (residues 336-518) adapts an open up-conformational state [65]. Peptidase domain of hACE2 clashes when all the RBD domains of the homotrimer SARS-CoV-2 S are in down conformational state. When compared with the SARS-CoV-1 RBD, an hACE2-binding ridge in SARS-CoV-2 has more compact conformation and several changes in the interface residues which stabilize two virus-binding hotspots at the RBD-hACE2 interface [64].

Interface interactions certainly play crucial role for binding and dynamics of domain motion. Initially, we use the Prodigy webserver to evaluate these interactions [66]. These calculations indicate that interface between CoV-2-RBD and hACE2 is relatively larger than the SARS-CoV-1 complex, with higher potential for large intermolecular interactions. The interface properties of the complexes are listed in Table 1. In both complexes, almost all interface molecular interactions involve loops and small parts of beta-sheets secondary structures of RBD and a single alpha helical structure of hACE2, indicating interface instability (Figure 1). We find uneven distribution of interface interactions along this single hACE2 alpha-helix. While the two end portions and the middle part of the helix hold most interactions, there are other interacting residues between CoV-1 and CoV-2 complex (see Appendix A). In particular, we note that hACE2.GLU35[OE2]–RBD.GLN493[NE2] and hACE2.THR27[O]–RBD.TYR489[OH] come closer (2.6 Å) and form H-bonds in the CoV-2-RBD-hACE2 complex (Figure 1). Inclusion of these two additional polar interactions may affect subsequent proteolytic processing of S protein and membrane fusion with the S2 unit. The Prodigy treatment yields almost similar binding free energy for both complexes at T = 25 °C. The latter contradicts the experimental observations, for example, recent measurements by surface plasmon resonance at T = 37 °C provide evidence [67] that the binding energy is lower for the CoV-2-hACE2 complex than that for CoV-1-hACE2 complex. These experiments estimate the difference ΔG between the relevant binding energies as $\Delta G_{exp}=-1.1$ kcal/mol.

In order to check the reliability of the predicted interface interactions, we have calculated ^13^C chemical shifts (δ) using structural coordinates of both the CoV-1and CoV-2 hACE2 bound complex. This calculation is carried out using SHIFTX2 [70] which combines ensemble machine learning approaches with the sequence alignment-based methods. We note that there is no overall significant differences in δ (ppm) between the hACE2 bound CoV-1and CoV-2 complex (Figure 2A). However, relative absolute differences Δδ are prominent at the interface between hACE2 and RBD (Figure 2A middle). Detected uneven distribution and changes of interface interactions are shown to be strongly correlated to deviation in the chemical shifts (Δδ). In particular, the hACE2 interface helix and the interacting RBD domain of the CoV-2 RBD shows significant difference in Δδ (Figure 2B). As expected, very high Δδ values are noted at and nearby H-bond forming residues GLU 35, THR 27 on the hACE2 helices and GLN 493 and TYR 489 on the RBD interface (Figure 2B). A large Δδ pick of 14.86 ppm on the RBD-interface and 2.50 ppm on the hACE2 helix are noted, with significant changes in the middle and end parts of the interacting interface helix. Furthermore, large picks in Δδ corresponds to AA substitutions. For example, Δδ pick of 14.86 and 8.90 ppm correspond to the substitution THR433GLY446 and THR487ASN501 on the interacting RBD domain.

### 2.2. 3DRISM Studies of Water-Mediated Interactions in SARS-CoV-1-hACE2 and SARS-CoV-2-hACE2 Complexes

As we have seen, the preliminary bioinformatics treatment reveals an essential role of interfacial interactions in the formation of RBD-hACE2 complexes. However, the simple Prodigy analysis can not evaluate the relative stability of SARS-CoV-hACE2 complexes. To provide further insight into the problem, we perform the 3DRISM treatment (see Section 4). Figure 3A demonstrates a water distribution in the interfacial region for CoV-2-hACE2 complex. Water molecules are depicted only in the most probable positions. As it is seen, the interfacial water interacts strongly with the RBD as well as with the hACE2 subunits, the latter confirms the previous molecular dynamic simulations [71]. However, our calculations yield the water-mediated interactions to be stronger for the CoV-2-hACE2 complex than those for the CoV-1-hACE2. See for example Figure 3B demonstrating a relative difference in distributions of water oxygens (blue) and hydrogens (red) between the two complexes. Moreover, the water interacts more strongly for CoV-2-hACE2 rather than for pure coronavirus (see Appendix A). To confirm this effect we calculated protein-solvent interaction energies (Table 1). Although the Lennard-Jones energy between the CoV-1-hACE2 and water is stronger than that for the CoV-2-hACE2 complex, the total interaction energy has a more profit for the latter complex due to a strong decrease in the electrostatic energy. This observed effect can be explained by a polarization of interfacial water which is stronger for the CoV-2-hACE2 complex. Apart from it, the latter complex forms water bridges whose number is larger by 2 than in the case of CoV-1-hACE2. At the same time the water bridges is stronger in the CoV-2-hACE2 aggregate. To prove it, we have calculated the potential of mean force (pmf) between atoms of RBD and hACE2 receptor (Figure 4). It is clearly seen that bridging water molecule forms simultaneously hydrogen bonds with CoV-2 (Gln493) and hACE2 (Glu35) amino acids in the case of CoV-2. However, the similar hydrogen bond with bridging water is very weak and diffusive for the CoV-1-hACE2 complex (Figure 4). Therefore, we conclude that bridge water molecules play a significant and, perhaps, crucial role in stabilizing CoV-2-hACE2 complex.

We have also calculated the binding energies for the complexes under consideration. For this purpose, we split the RBD and hACE2 parts of the complexes and calculated them separately. The binding energy is evaluated as a difference between the sum of free energies of the subunits and that for the whole complex. The results are indicated in Table 1. We note that the absolute values of the calculated energies are varied as calculation box size changes, nevertheless the relative difference ΔG between the binding energies remains the same ΔG=−7.1 kcal/mol. The latter is in an agreement with results obtained by coarse-grained (CG) simulations $\Delta G_{CG}=-4.3$ kcal/mol [69] and with the use of molecular mechanics/GBSA calculations [68] yielding $\Delta G_{GBSA}=-14.9$ kcal/mol. Simple 3DRISM analysis of contributions to the binding affinity indicates that the major difference come from electrostatic contributions. The latter correlates also with our estimation of Lennard-Jones and Coulomb parts in protein-water interactions. We note all the indicated methods seem to underestimate the difference in the binding energies with respect to the experimental data. The reason of this drawback seems to be ionic effects which can reduce the electrostatic contribution as well as entropic changes caused by relaxation of individual subunits. We note also that all the methods provide absolute values of the binding energy varying by several times. Moreover, detailed analysis of an influence of conformational changes [68] on the binding demonstrates that the energy and hence binding affinity depends strongly on a conformation state of the complexes, conformation angle between RDB and hACE2, etc. Therefore, additional analysis of these effects is to be required to prove the 3DRISM findings.

### 2.3. Posterior Bioinformatics Treatment by NMA and CGS

Significant conformational changes are commonly reflected in the differential domain motions. To indicates these changes we draw the Ramachandran plots for the complexes. They display large changes in phi and psi dihedral angles for most of the RBD residues in favorable and allowed regions of the plot (Figure 5B). These analyses suggest characteristic conformational differences in hACE2-bound and non-bound RBD between SARS-CoV-1 and SARS-CoV-2 and, therefore, it necessitates further examination of its functional significance and rigidification properties of the all potential RBD conformations. To capture such differences, we have conducted NMA using the Gaussian network model and the anisotropic-network model, utilized in DynOmics webserver [72] and in iMODS [73]. We calculated the first twenty slowest modes for all the CoVs structures. The eigenvectors of these modes represent the global motions and the constrained residues help in identifying critical regions such as hinge-bending regions and thereby giving an idea of domain motions around these regions. This hybrid ENM has efficiently captured the dynamic differences between SARS-CoV-1 and SARS-CoV-2 with and without hACE2 bound. With the trimeric CoV-1 and CoV-2 macromolecular structure, we have calculated covariance matrix in iMODS which is computed using the Cartesian coordinates and the Karplus equation of collective motion in protein. This covariance matrix signifies coupling between pairs of residues. Overall, covariance patterns are very similar for CoV-1and CoV-2; however, there are a very few sharp differences in few spots of the covariance matrices. As usual, such differences are prominent in the low-frequency normal modes. To examine the internal residual coupling and its effects, we have studied low-frequency normal modes for RBD-up forms of both the structures and noted that the second normal mode is capable to effectively capture these differences. It shows that RBD has relatively high amplitude motions for both the CoVs without hACE2 binding, while the S2-unit holds lower-amplitude motions (Figure 5A). When we compare this prefusion up-form RBD local motion along the residues of CoV-1and CoV-2, it shows that the extended RBD region follows anti-phase like dynamics; CoV-2-RBD has positive eigenvector components in oppose to negative components of CoV-1-RBD (Figure 6B). In contrast, these anti-phase dynamics are mostly vanished due to hACE2 binding; hACE2 binding makes the RBD relatively stable, with little faster movements of CoV-2-RBD (Figure 5A and Figure 6A).

We applied the computational method FRODAN [74,75] (see Section 4) on the whole spike protein with the RBD in the up-state and the RBD-hACE2 complex of the single RBD domain bound to the hACE2. The method (see Figure 7) reveals RMSF fluctuations of the respective backbone atoms in both spike proteins at different hydrogen-bond energy cut-offs (temperatures). Consistent with the normal mode analysis, it is evident that for both spike proteins in SARS-CoV-1 and SARS-CoV-2, the RBD in the up conformation has higher conformational fluctuation relative to other domains. As we increase the hydrogen-bond energy cutoff (i.e., removing weak transient hydrogen bonds) conformational fluctuations in RBD tend to further increase. There is an overall increase in conformational dynamics in SARS-CoV-1 S structure in comparison to the SARS-CoV-2 (Appendix A). In particular, RBD has higher RMSF in SARS-CoV-1. This is in agreement with our rigidity analysis (see details below), where we have shown that SARS-CoV-2 retains its overall rigidity better than SARS-CoV-1. Furthermore, large-scale computing efforts via Folding@home project have carried out millisecond MD simulation [76], where it was shown the RBD domain in the up state of the SARS-CoV-1 exhibits higher deviation from the respective crystal structure in comparison to the SARS-CoV-2 [76]. Interestingly, one of the RBDs in SARS-CoV-1 that is initially in the down configuration transitions to the open configuration. This correlates with previous studies which have shown that for the SARS-CoV-1, the two-up and three-up RBD states are also populated in the unbound spike protein [61]. On the other hand, for SARS-CoV-2 previous studies indicated that the two-up and three-up states are rarely observed [77]. This trend is reflected also in our simulation results. For SARS-CoV-2, the RBDs that are initially in the down state remain in the closed state at wide range of energy cut-offs. Overall, it is evident that temperature increase affects the SARS-CoV-1 spike protein’s conformational stability in the more pronounced way than the SARS-CoV-2. This difference may provide clues why RBD binds tighter with hACE2 in SARS-CoV-2, and a general trend that SARS-CoV-1 is more sensitive to environment conditions than SARS-CoV-2 [64,78,79].

Figure 7 shows RMSF profiles of RBD and hACE2 in the complex at different energy cut-offs. Similarly, Similarly, as in the whole spike protein, the increase in the hydrogen-bond energy cutoff leads to the overall higher RMSF values in both complexes. The RBD domain fluctuates less in SARS-CoV-2. The average RMSF in SARS-CoV-2 is lower at all considered cut-offs (Appendix A). This is supported by rigidity analysis findings that are indicating excessive flexibility in the SARS-CoV-1-hACE2 complex. Overall, it is evident that the stability of SARS-CoV-2-hACE2 complex results from the stronger interface contacts [64]. On the other hand, with the increase in the hydrogen-bond energy cutoff, hACE2 fluctuation magnitude increases in the similar manner in both SARS-CoV-1 and SARS-CoV-2 (Appendix A).

To provide further insight to structural flexibility, we have carried out rigidity analysis using Floppy Inclusion and Rigid Substructure Topography (FIRST) program [80], which is based on the pebble game algorithm and techniques in mathematical rigidity theory [81]. We used FIRST to decompose the spike protein structures into rigid clusters with the input of H-bond energy cutoff (H-cut-off, kcal/mol). With H-cut-off of −0.1, −0.5, −1.0, −1.5, −2.0, −2.5, and −3.0, CoV-1 and CoV-2 spike with- and without hACE2 bound have been decomposed. Number of rigid clusters and degree of freedoms consistently increase with increasing H-cutoff. Without hACE2 bound, CoV-1 S have been decomposed into higher number of rigid clusters with much higher degrees of freedom relative to CoV-2 S (see Appendix A). At a low energy cutoff (−0.5 kcal/mol) SARS-CoV-2 is significantly more rigid than SARS-CoV-1, as it is dominated by a few large rigid clusters. As we increase the energy cutoff, SARS-CoV-1 continues to gain more flexibility while SARS-CoV-2 retains most of its rigidity. At −1.5 kcal/mol SARS-CoV-1, including its RBD, has lost almost all internal structural rigidity, while SARS-CoV-2 still maintains significant rigidity. Extending this analysis to the whole complex, it is evident that the SARS-CoV-2 RBD-hACE2 complex at −1 kcal/mol is dominated by one large rigid cluster, whereas SARS-CoV-1 RBD-hACE2 complex is more flexible consisting of several rigid cluster. As hydrogen bond energy cutoff is increased, SARS-CoV-1 RBD-hACE2 complex is losing its rigidity more rapidly than SARS-CoV-2 RBD-hACE2 complex. This strongly indicates that CoV-1 S becomes more flexible, generating a large conformational ensemble with less potential for binding with hACE2. This result is consistent with the recent observations in an extensive 0.1 s MD simulation which noted a large opening of spike with presence multiple cryptic epitopes [76]. In contrast, when RBD binds with hACE2, CoV-2-RBD-hACE2 becomes relatively more rigid than CoV-1-RBD-hACE2, which favors proteolytic processing for membrane fusions.

## 3. Discussion and Conclusions

Using recently reported cryo-EM structures of SARS-CoV-1 and CoV-2 S protein we have investigated binding and conformational properties of the CoV-hACE2 by utilizing a combination of SDFT and bioinformatics methods. The SDFT has been applied within the framework of 3DRISM. The method is used to reveal the role of water in interfacial interactions between RBD and hACE2 and to evaluate the difference in the binding affinities for the complexes under the consideration. The normal mode analysis, constrained geometric simulations, and the rigidity analysis have been used to analyze a role of flexibility and conformational motion on the stability of the CoV-hACE2 complexes. The 3DRISM analysis reveals the essential role of interfacial water for the complex stability. It indicates the CoV-2-hAC2 is more stable than CoV-1-hAC2 complex and has an order of magnitude lower dissociation constant than the CoV-1-hAC2 complex. The latter is in an quantitative and qualitative agreement with the experimental data and other calculations based on the CG and the MM/GBSA methods.

We note that the considered SDFT may be very useful to treat stability of coronavirus binding to hACE2, especially when the method is accompanied by further bioinformatics analysis for treating flexibility and conformational motions. To provide such analysis we have used the the NMA, CGS, and FIRST tools. Using these treatments we indicate that the RBD-up conformation in the SARS-CoV-1 is less stable than in the SARS-CoV-2. When bound with higher surface area of CoV-2 RBD and large local conformational changes in the AA residues, it establishes higher interactions with the human receptor hACE2. These underlying conformational differences are illustrated with the higher RMSD of C*α* atoms, changes in phi and psi dihedral angles and relatively large amplitude anti-phase-like RBD motion. In particular, we note that hACE2.GLU35–RBD.GLN493 and hACE2.THR27–RBD.TYR489 come closer in the CoV-2-RBD-hACE2 complex and forms important H-bonding interactions. However, it remains to be explored how efficiently CoV-2 regulate and open-up RBD for hACE2-binding. The RBD of CoV-2 forms stronger water bridges with the hACE2, and it plays a significant role in the total stabilization of theCoV-2/hACE2 complex. Inclusion of the two noted additional polar interactions affects structural flexibilities. We find such interaction changes critically affect structural flexibility. In particular, hACE2 binding makes the CoV-2 structure more rigid relative to CoV-1complex. These hACE2 induced changes in structural flexibility favours subsequent proteolytic processing which is essential for membrane fusion. Understanding prefusion conformational dynamics as well as its binding mechanism to the receptor among closely related species is critical for designing vaccine and inhibitors to strop viral entry. Recent major therapeutic efforts are targeting the interactions between the SARS-CoV-2-S and the hACE2 [82,83]. This understanding certainly provides a clue for designing novel vaccine and antiviral drugs.

## 4. Methods

### 4.1. Classical Site Density Functional Theory

We investigate hydrated protein complexes. Although there are numerous data about an essential role of ions in stabilization of protein complexes [84,85], we ignore this effect at the current level of the consideration and treat the solvent as a pure water. Then, the problem reduces to evaluations of site density profiles $\rho_{i}\left(r\right)\;\left(i=O,H\right)$ of inhomogeneous water subjected to external potential v(r) caused by solvent-protein interactions. As we indicated earlier [22,23,86], the rigorous classical SDFT formulation is to be based on construction of a generating functional depending on two coupled variables site densities $\rho_{i}\left(r\right)$ and site field $J_{i}\left(r\right)$ and further evaluations of the functional. The densities and fields are to be obtained by the relevant minimization of the functional. Omitting all the technical details which have been published earlier [22,23,86], we provide here only the final relations for the fields and densities. In the vector form, they can be written as
(3)$$\left\{\begin{array}{c} \rho \left(r\right)=\rho_{0}\left[1+\xi ([J],r)\right]e^{-\beta J(r)}\\ J(r)=v(r)+\phi (r) \end{array}\right.$$

The meaning of the relations is rather simple. The first equation represents the density of an inhomogeneous molecular liquid subjected to the field J(r), while the second one closes the self-consistent loop and indicates the field to be composed from the solvent-protein potential v(r) and intermolecular potential ϕ(r) caused by interactions between solvent molecules. The first relation looks like as an barometric expression, but it contains an additional term ξ([J],r) referred to as the correlation hole functional, which accounts for intra-molecular correlation effects. All variety of the SDFT models lies in expressions for the correlation hole functional and intermolecular potential. In the general case the latter can be presented as a density functional:(4)$$\begin{array}{cc} -\beta \phi (r) & =\int \left(S_{m}^{-1}(|r-r^{\prime }\left|\right)-S^{-1}(|r-r^{\prime }\left|\right)\right)\Delta \rho \left(r^{\prime }\right)dr^{\prime }+b\left(\rho \left(r\right)\right) \end{array}$$
where $S_{m}\left(r\right)$ and S(r) is a structure factor of single water molecule and uniform water, respectively, while b(ρ(r)) is the bridge functional accounting contributions beyond the linear response. Various approximations can be used for this functional and its distance dependence can be extracted from molecular simulations [87,88,89].

The presence of the correlation hole functional reveals the main difference between treatment of molecular and simple liquids. The functional expresses the fact that the external field imposed on a given site will be also propagated to all other sites through the molecular bonds. The correlation hole functional can be expressed as a cluster expansion in terms of the Mayer function $f\left(r\right)=e^{-\beta J(r)}-1$ and intramolecular correlation functions, $D^{\left(s\right)}$:(5)$$\begin{array}{c} \xi \left(\left[J\right],r\right)=\sum_{s=2}^{M}\frac{\mathrm{Tr}[D^{\left(s\right)}f^{s-1}]}{(s-1)!} \end{array}$$
where *M* is the total number of water sites. We note that the second order intramolecular correlation function $D^{\left(2\right)}\left(r\right)$ depends only on bond lengths $l_{ij}$, whereas the third order one involves angles between bonds, the forth one depend on dihedral angles, etc. As a result, the expansion for liquids whose molecules containing more than two sites requires integration over angles. Therefore, the SDFT application to water models like as SPC (simple point charge) model is a time consuming procedure.

The 3DRISM is an alternative to the rigorous SDFT treatment. The model assumes that correlation hole depends only on bond lengths and can be expressed as [22]:(6)$$\begin{array}{cc} \xi (r) & \approx \xi^{RISM}\left(r\right)\equiv \exp\left[\int (1-S_{m}^{-1}(|r-r^{\prime }|))\Delta \rho \left(r^{\prime }\right)dr^{\prime }\right]-1 \end{array}$$

As a result, the relation between the site fields and densities are rather simplified. If we introduce new variable referred to as site direct correlation function c(r):(7)$$\begin{array}{c} c\left(r\right)\equiv \int S^{-1}(|r-r^{\prime }|)\Delta \rho \left(r^{\prime }\right)dr^{\prime } \end{array}$$

Then the 3DRISM relations can be rewritten in the form similar to those in simple fluids:(8)$$\begin{array}{ccc} \rho (r) & = & \rho_{0}e^{-\beta v\left(r\right)+\frac{\Delta \rho (r)}{\rho_{0}}+c\left(r\right)+b\left(r\right)}\\ \Delta \rho (r) & = & \rho_{0}\int S(|r-r^{\prime }\left|\right)c\left(r^{\prime }\right)dr^{\prime } \end{array}$$

The first one of these relations is referred to as a closure, while the second one is the site Ornstein-Zernike integral equations. Due to neglection angular dependencies in intramolecular correlations, the computational 3DRISM costs are by two orders less expensive than the SDFT calculations by (Equation 3)–(Equation 5). The 3DRISM input is protein-solvent potential v(r), susceptibility of pure water S(r), bridge function (or functional) b(r), and parameters determining thermodynamic state of water (i.e., *T* and $\rho_{0}$). It is important that calculating site densities by Equation (Equation 8) we fix a configuration of protein complex. In general, we can also calculate not only site densities but also changes in free-energy of the complex caused by hydration. Various approximation can be used for the free energy [33], we will use the so-called the pressure correction approximation which has been shown [48] to provide rather accurate evaluations for the hydration free energy.

We note the complexes under our consideration are too large to be studied by the 3DRISM in the fully atomistic format. For example, the size of 6M17 complex exceeds 300 Å, while the number of heavy atoms in this complex is more than 24 thousands. It is obvious that only residues near the interface region yield an essential contribution to the binding, whereas others located at large distances provide only external electrostatic field affected the interface region. To select the regions of with the essential contribution, we apply an iterative scheme. First, using the data obtained from the bioinformatics treatment we consider interfacial region which includes all contacts indicated in Appendix A. Then we extend the size of calculation box and recalculate the water distribution. The iterations are repeated until the changes in the calculated water distribution, caused by the extension of the calculation box, become less than $10^{-3}$ in absolute values. The final clipped regions are shown in Appendix A. Details of the iteration scheme are given in Appendix A. The similar consideration has been carried out for SARS-CoV-1 (pdb id: 6CRZ) and SARS-CoV-2 (pdb id: 6VYB) complexes (see details in the Supporting Information). The 3DRISM with the 3D-Kovalenko-Hirata closure was used to calculate these complexes. The calculations were carried out for the studied complexes hydrated in water at ambient conditions. For water the modified version of the SPC/E model (MSPC/E) was used [90]. The corresponding LJ parameters of the solute atoms were taken from the ff14SB force fields [91]. The 3DRISM equations were solved on a 3D-grid of 350 × 320 × 350 points with a spacing of 0.025 nm. A residual tolerance of $10^{-6}$ was chosen. These parameters are enough to accommodate the complex together with sufficient solvation space around it so that the obtained results are without significant numerical errors.

### 4.2. Bioinformatics Tools

#### 4.2.1. Root-Mean-Squared Deviation (RMSD) of C-*α* Atoms

We used MatchMaker in UCSF-Chimera to superimpose structures. It performs a best fit after automatically identifying pair of residues. It considers similar structures to be superimposed while there are large sequence dissimilarities by producing residue pairing uses with the inputs of AA sequences and secondary structural elements. After superposition, it uses the usual Euclidian distance formula for calculating RMSD in angstrom unit with the use of PDB atomic coordinates files.

#### 4.2.2. Normal Mode Analysis

DynOmics ENM Server [72] was used to perform normal mode analysis (NMA). NMA is a well-explored technique for exploring functional motions of proteins. It combines two elastic network models (ENMs)—the Gaussian Network Model (GNM) and the Anisotropic Network Model (ANM) to evaluate the dynamics of structurally resolved systems, from individual molecules to large complexes and assemblies, in the context of their physiological environment. In the GNM model, network nodes are the C-alpha atoms and the elastic springs represented the interactions. We used GNM with interaction cut-off distance of 7.3 Å and spring constant scaling factor cut-off of 1 Å for the calculation of the elastic network model. We calculated the first twenty slowest modes for all the CoVs structures. The eigenvectors of these modes represent the global motions and the constrained residues help in identifying critical regions such as hinge-bending regions and thereby giving an idea of domain motions around these regions. We plotted the second slowest mode in different conditions which showed a significant difference in motions.

#### 4.2.3. Rigidity Analysis

We use the program Floppy Inclusion and Rigid Substructure Topography (FIRST) [80]. In this approach, we first create a constraint network, where the spike protein is modeled in terms of nodes (atoms) and edges (covalent bonds, hydrogen bonds, etc.). A hydrogen bond cutoff energy value is selected where all bonds weaker than this cutoff are ignored. The strength of hydrogen bonds is calculated using a Mayo energy potential. The spike protein network is next decomposed into rigid clusters and flexible regions. The value of energy strength was selected in such way that the bonds strength below this cut off were ignored. First then applies the rigorous mathematical theory of rigid and flexible molecular structure and pebble game algorithm calculates the conformational degrees of freedom rapidly decompose a protein into rigid clusters and flexible region.

#### 4.2.4. Constrained Geometric Simulations for Conformational Sampling

To further probe the dynamical features of the whole spike protein and the impact of hACE2 binding, we have applied the CGS by the Framework Rigidity Optimized Dynamics Algorithm New (FRODAN) [74,75]. This method utilizes a coarse-grained molecular mechanics potential based on rigidity theory to explore receptor conformations well outside the starting structure. FRODAN can be regarded as a low computational complexity alternative to MD simulations which can sample wide regions of high dimensional conformational space. We run FRODAN in the non-targeted mode, generating 30,000 candidate structures for each case. The temperature impact was evaluated by running simulations at different hydrogen bond energy cut-offs. The considered range of the energy cut-offs was from −1.0 to −3.0 kcal/mol, where the more negative cut-off values correspond to higher temperature. During each individual run the cut-off value was kept constant. Before running simulations, hydrogen bonds to each considered structure were added using MolProbity server [92].

## Figures and Tables

**Figure 1 molecules-27-00799-f001:**
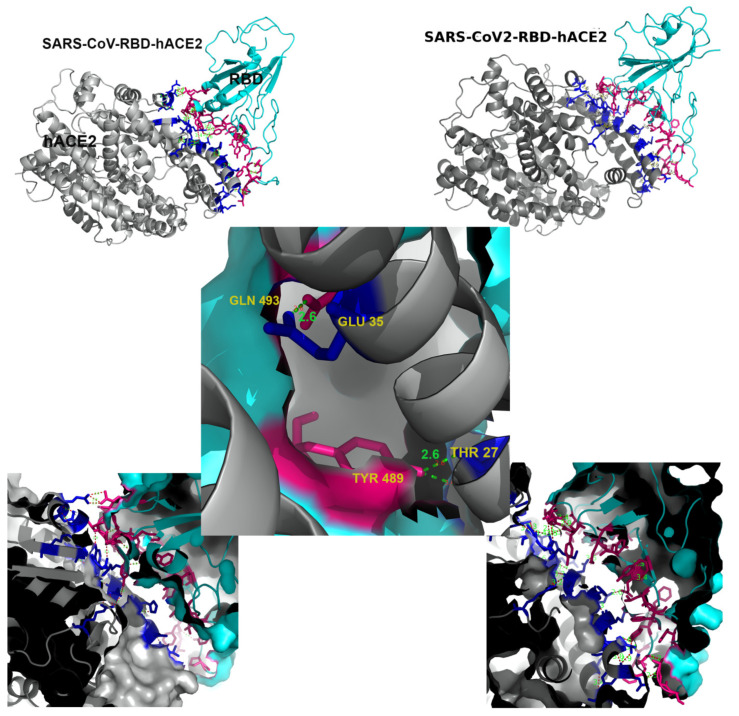
Interface interactions between CoV-1 and CoV-2 receptor-binding domain (RBD) and the hACE2. In both complexes, almost all interface molecular interactions involve loops and small parts of beta-sheets secondary structures of RBD and a single alpha helical structure of hACE2. The hACE2.GLU35–RBD.GLN493 and hACE2.THR27–RBD.TYR489 (2.6 Å) form H-bond interactions indicated by dashed lines. Pink and blue color represent RBD and hACE2 interfaces respectively.

**Figure 2 molecules-27-00799-f002:**
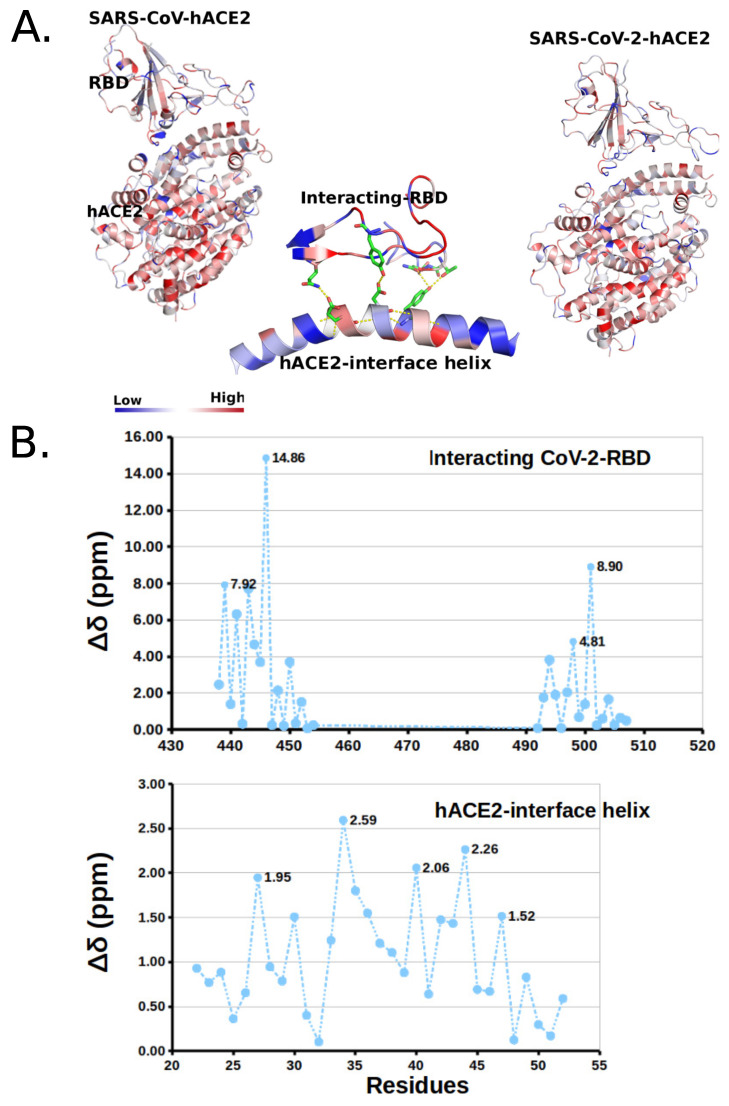
^13^C Chemical shifts δ caused by interface interactions between hACE2 and RBD. (**A**) Changes in δ for the complexes (top two figures), the middle figure shows the deviation in chemical shifts (Δδ) at the interface interacting sites. (**B**) Change in Δδ observed at H-bond forming residues GLU 35, THR 27 on the hACE2 helices and GLN 493 and TYR 489 on RBD interface.

**Figure 3 molecules-27-00799-f003:**
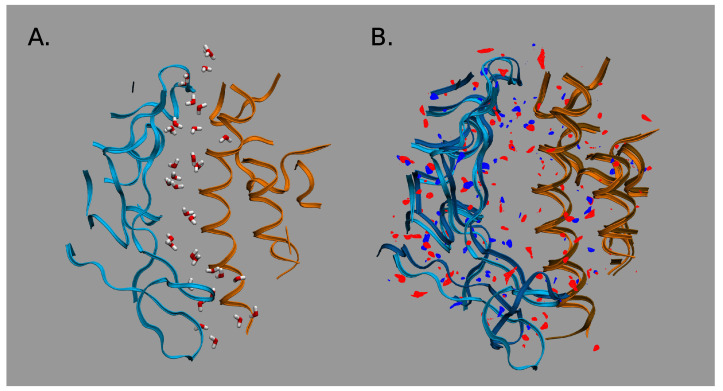
Water distribution in the interfacial region of complexes: (**A**) For the CoV-2-hACE2 complex. (**B**) Differences in distributions of water oxygens (blue colour) and water hydrogens (red colour) between the CoV-2-hACE2 and CoV-hACE2 complexes. The RBD is indicated by blue ribbons, the hACE2 by orange ribbons, and the CoV-hACE2 is shown as background for the differences.

**Figure 4 molecules-27-00799-f004:**
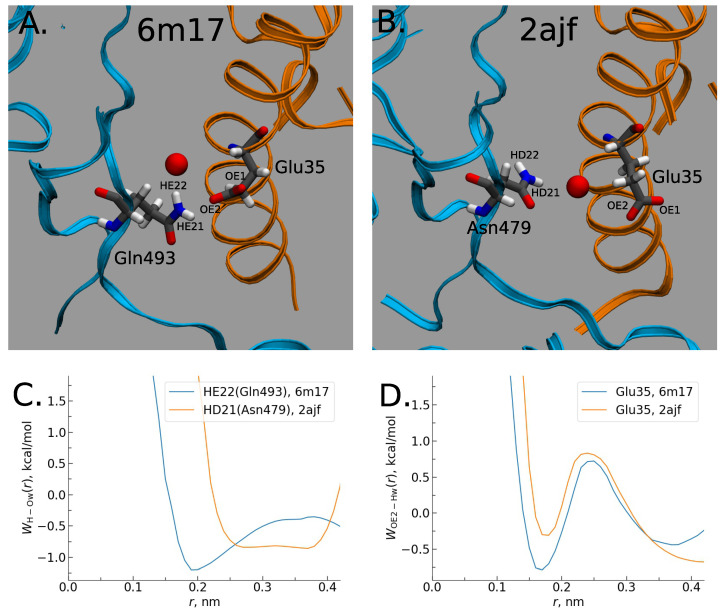
Water bridging the complexes: (**A**) Location of water oxygen bridging the CoV-2-hACE2 complex. (**B**) The same for the CoV-1-hACE2 complex. (**C**) The pmf of H-O distribution for the CoV-2 and CoV-water bridges. (**D**) The pmf of O-H for the hACE2-water bridges in the complexes.

**Figure 5 molecules-27-00799-f005:**
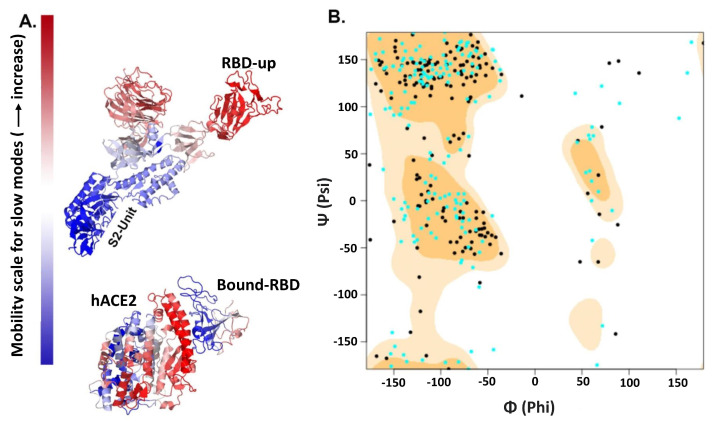
NMA (**A**) and Ramachandran plot (**B**) for RBD-hACE2 complexes. Cyan dots represent CoV-2-RBD AA residue phi-psi position, while black is for CoV-1 complex. We used low-frequency second normal mode to differentiate between SARS-CoV-1 and SARS-CoV-2 with and without hACE binding. (**A**) Its RBD-up conformation region in the S1-unit has higher mobility relative to other parts of the S-protein. In contrast, hACE2 binding make the RBD less mobile. (**B**) Ramachandran plot of AA residues of RBD bound with hACE2 shows significant conformational changes in the RBD.

**Figure 6 molecules-27-00799-f006:**
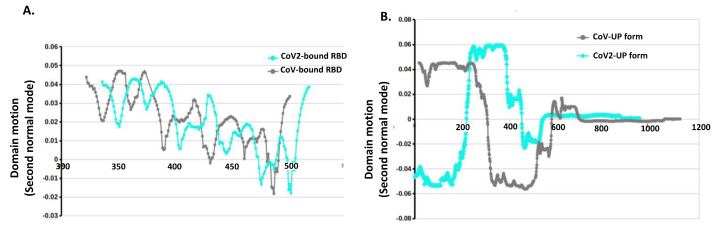
Domain motions in second normal mode (**A**) Residue-wise eigenvectors of bound RBD AA residues is representing low-frequency motions. There are no significant differences in bound RBD motions between both the SARS-CoV-1 complexes. (**B**) Without hACE binding, RBD up-conformation protomer is showing significant differences in the domain motion in the RBD regions. It is like anti-phase-type dynamics between the two CoVs structures. Cyan dots represent CoV-2-RBD AA residue phi-psi position, while black is for CoV-1 complex.

**Figure 7 molecules-27-00799-f007:**
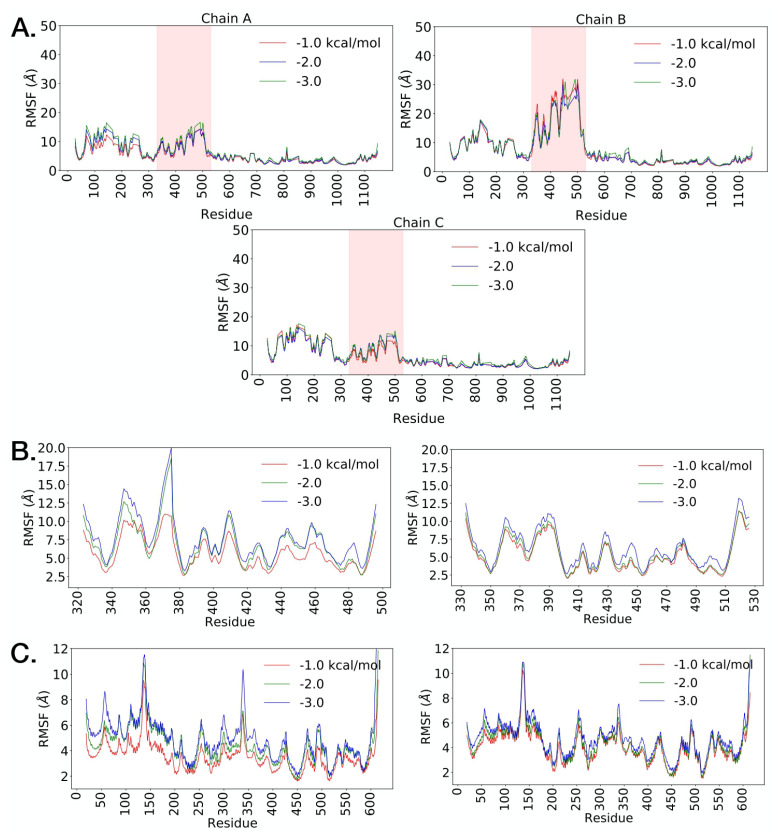
Conformational dynamics analysis using FRODAN method: (**A**) Backbone RMSF profiles SARS-CoV-2 Spike protein with RBD shown in pink. (**B**) RMSF of the RBD domain in the complex with hACE2 (left SARS-CoV-1, right SARS-CoV-2). (**C**) RMSF of the hACE2 domain in the complex (left SARS-CoV-1, right SARS-CoV-2).

**Table 1 molecules-27-00799-t001:** Interface properties of RBD-hACE2 complexes for CoV-1 and CoV-2.

Parameter	CoV-1-RBD-hACE2	CoV-2-RBD-hACE2
structure		
Bound-RBD surface area (Å^2^)	18,435.6	19,866.4
Interface area (Å^2^)	796.7	827.7
thermodynamics		
Binding Energy (kcal/mol)		
PRODIGY	−11.1	−10.8
EXP [67]	−10.7	−11.8
MM/GBSA [68]	−10.0	−24.9
CG [69]	−66.4	−70.7
3DRISM	−50.1	−57.2
*U_LJ_* (kcal/mol) *^a^*	−281	−261
*U_C_* (kcal/mol) *^a^*	−4887	−5160

^*a*^*U_LJ_* and *U_C_*—Lennard-Jones and Coulomb parts of complex-water interactions.

## Data Availability

Not applicable.

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
