# Peer review of "Site Density Functional Theory and Structural Bioinformatics Analysis of the SARS-CoV Spike Protein and hACE2 Complex"

_molecules, 2022, doi:10.3390/molecules27030799_

Round 1

Reviewer 1 Report

Dear Authors,

The work is beautifully organized  and presented.It would be benefical to revise your writing and figures by reviewing the again. 

With this aspect, the study has a very interesting and original point of view.

In the study, combined computational methods are used together. The transmission of the disease has been evaluated at both atomic and molecular level. The work is well written and understandable.

The obtained data should be visualized in a consistent but somewhat simplified way, and its evaluation will make the study more effective.

They address the main question being asked, proofs have been made to support it from beginning to end.

Regards,

Author Response

Thank you very much for recognizing our contributions and pointing out the novelty of our work and its significance. We have further revised the manuscript by separating the results and method section. The method section now contained detailed on SDFT-based computation and all the structural bioinformatics approaches used in thus study. Figures are revised with detailed captions and larger font size. English grammatical errors were also removed in the revised version.

Reviewer 2 Report

In this manuscript, the authors reported a study that integrates classical SDFT and bioinformatics tools to investigate the complex formed between RBD domain of the S protein in SARS-CoV and human ACE2 receptor via simulations. The differences between the complexes formed from SARS-CoV-1 and SARS-CoV-2 were identified using a number of computational methods. The study also highlighted the role of water molecules in mediating the formation of the complex.

Although the manuscript presents interesting insights into the mechanistic basis of the more stable complex formed from SARS-CoV-2 S protein compared to SARS-CoV-1 and clear rationale for using the computational methods chosen, it has low readability in its current format. I suggest that the authors rewrite the manuscript to improve scholarly representation. I have some specific suggestions for making improvement.

  1. The authors need to have a separate section that contains description of methods rather than mixing up results and methods. The lengthy description of methods while presenting results prevent the readers from finding the main results and conclusions easily. Given the large number of methods/software packages used by this study, a new method section that contains subsections, each of which devoted to the description of one or several related methods should be included.
  2. The quality of figures needs to be improved. For example, in Figure 8, the lines are too faint to be seen clearly. Please also check the sizes of axes labels in all figures to ensure they are not too small.
  3. Many grammatical mistakes and inaccuracy hinder the understanding of this manuscript. Please pay attention to the use of the appropriate adjectives. For example, in “interactions are higher”, higher is not the right word. Stronger is what the authors should have used. In numerous places, “later” should be replaced with “latter”.

Below are additional specific suggestions.

Figure 7. The figure caption should explain the color scheme used in (B), namely what the cyan and black dots stand for. The authors tried to show large differences in phi & psai angles in Figure 7B. Although the difference is obvious, the authors did not explain clearly what the differences real mean in terms of differences in secondary structure. Did any part of the protein change its secondary structure? This information is more critical than the distribution the phi & psai angles for the entire protein. For the Ramachandran plot, it is not possible to deduce what part(s) of the protein changed most significantly, since the changes were not plotted against residue number. I suggest that the authors improve their presentation of data and draw a clearer conclusion.

Figure 8. Y axis labels needed to be enlarged for both panels. They are barely legible. The X axis needs to be drawn in black line (rather than the grey which is in distinguishable from the grids).

Line 331-333, where are the sharp differences shown?

Line 60, “later” should be replaced with “latter”

Line 88, “later” should be replaced with “latter”

Line 193,”higher” should be replaced with “larger” or alternatively the authors should state that the interface area is higher rather than “interface … higher”.

Line 275, remove “more” from “more stronger”

Author Response

Reviewer’s #2 comments-1

In this manuscript, the authors reported a study that integrates classical SDFT and bioinformatics tools to investigate the complex formed between RBD domain of the S protein in SARS-COV and human ACE2 receptor via simulations. The differences between the complexes formed from SARS-CoV-1 and SARS-CoV-2 were identified using a number of computational methods. The study also highlighted the role of water molecules in mediating the formation of the complex.

Reply: Thank you very much for carefully reading our manuscript and for identifying the primary problem that has been tackled by the research team.

Reviewer’s #2 comments-2

Although the manuscript presents interesting insights into the mechanistic basis of the more stable complex formed from SARS-CoV-2 S protein compared to SARS-CoV-1 and clear rationale for using the computational methods chosen, it has low readability in its current format. I suggest that the authors rewrite the manuscript to improve scholarly representation.

Reply: Thank you for recognizing our contributions. We have now thoroughly revised the manuscript by removing english grammatical errors, separating method section from the result, and improving the figure resolution with detailed captions.

Reviewer’s #2 comments-3

The authors need to have a separate section that contains description of methods rather than mixing up results and methods. The lengthy description of methods while presenting results prevent the readers from finding the main results and conclusions easily. Given the large number of methods/software packages used this study, a new method section that contains subsections, each of which devoted to the description of one or several related methods should be included.

Reply: Thank you for pointing out this problem. We have carefully addressed it in the revised version of the manuscript, in which method section detached from the results, and each method description was included in separate subsection. For the details, please see the revised manuscript.

Reviewer’s #2 comments-4

The quality of figures needs to be improved. For example, in Figure 8, the lines are too faint to be seen clearly. Please also check the sizes of axes labels in all figures to ensure they are not too small.

Reply: Yes, we have revised all the figures, including figure resolution and detailed captions.

Reviewer’s #2 comments-5

Many grammatical mistakes and inaccuracy hinder the understanding of this manuscript. Please pay attention to the use of the appropriate adjectives. For example, in “interactions are higher” higher is not the right word. Stronger is what the authors should have used. In numerous places, “later” should be replaced with “latter”.

Reply: Yes, we have found out all the language mistakes and revised in the new version. Issues of appropriate adjective use have also been fixed.

Reviewer’s #2 specific comments-6

Figure 7. The figure caption should explain the color scheme used in (B), namely what cyan and black dots stand for. The authors tried to show large differences in phi and psi angles in Figure 7B. Although the difference is obvious, the authors did not explain clearly what the differences really mean in terms of differences in secondary structure. Did any part of the protein change its secondary structure? This information is more critical than the distribution of phi and psi angles of the entire protein. For the Ramachandran plot, it is not possible to deduce what part(s) of the protein changed most significantly, since the changes were not plotted against reside number. I suggest that authors improve their presentation of data and draw a clearer conclusion.

Figure 8. Y axis labels needed to be enlarged for both panels. They are barely legible. The X axis needs to be drawn in black line (rather than the grey which is distinguishable from the grids)

Line 331-333, where are the sharp differences shown?

Line 60, “later” should be replaced with “latter”.

Line 193, “higher” should be replaced with “larger” or alternatively the authors should state that the interface area is higher rather than “interface … higher”.

Line 275, remove “more” from “stronger”

Reply: Thank you very much for pointing out all the mistakes. We have found out all those and revised accordingly. All figures were improved and corresponding captions were detailed to improve its visibility and clarity.

Round 2

Reviewer 2 Report

The authors have addressed all my concerns. I recommend publication of this manuscript in its current format.